

# Assimilating monthly precipitation data in a paleoclimate data assimilation framework

Veronika Valler[1,2], Yuri Brugnara[1,2], Jörg Franke[1,2], and Stefan Brönnimann[1,2]

[1]Institute of Geography, University of Bern, Bern, Switzerland
[2]Oeschger Centre for Climate Change Research, University of Bern, Bern, Switzerland

**Correspondence:** Veronika Valler (veronika.valler@giub.unibe.ch)

**Abstract.** Data assimilation approaches such as the ensemble Kalman filter method have become an important technique for paleoclimatological reconstructions and reanalysis. Different sources of information from proxy records and documentary data to instrumental measurements were assimilated in previous studies to reconstruct past climate fields. However, precipitation reconstructions are often based on indirect sources (e.g., proxy records). Assimilating precipitation measurements is a challenging task because they have high uncertainties, often represent only a small region and generally do not follow Gaussian distribution. In this paper, a set of experiments are conducted to test the possibility of using information about precipitation in climate reconstruction with monthly resolution by assimilating monthly instrumental precipitation amounts or the number of wet days per month, solely or in addition to other climate variables such as temperature and sea level pressure, into an ensemble of climate model simulations. The skill of all variables (temperature, precipitation, sea-level pressure) improved over the pure model simulations when only monthly precipitation amounts were assimilated. Assimilating the number of wet days resulted in similar or better skill compared to assimilating precipitation amount. The experiments with different types of instrumental observations being assimilated indicate that precipitation data can be useful, particularly if no other variable is available from a given region. Overall the experiments show promising results because with the assimilation of precipitation information a new data source can be exploited for climate reconstructions. Especially the wet day records can become an important data source in future climate reconstructions because many existing records date several centuries back in time and are not limited by the availability of meteorological instruments.

## 1 Introduction

Precipitation is one of the key components of the climate system. Understanding its variability is fundamental due to its impact on the ecosystem and on human society. Instrumental observations are the main data source for studying its spatio-temporal variability. However, instrumental measurements are often insufficient because long time series are rarely available and their spatial distribution is rather sparse, especially further back in time. To examine the decadal variability of precipitation longer time series are needed. This is often done by analyzing proxy records, documentary data or model simulations.

Simulations suggest, for instance, that the tropical monsoon regions are characterized by the largest interannual variability of annual precipitation and the interannual variability exhibits significant changes on the multi-decadal scale (Yang and Jiang,





2015). Precipitation reconstructions are needed to validate such results. Their number increased in the recent years on all scales from local to large-scale. To reconstruct local millennial long hydroclimate variability, tree-ring series were used for example in southern-central England (Wilson et al., 2013), and in southern Scandinavia (Seftigen et al., 2017). Pauling et al. (2006) reconstructed a 500-year long seasonal precipitation field over Europe back to 16th century, by using instrumental measurements, documentary data, and proxy records. A reconstruction of Northern Hemispheric hydroclimate variability from

multi-proxy records and documentary data is available between the 9th and 20th century (Ljungqvist et al., 2016). A similar reconstruction was also produced for southern South America for the last 500 years (Neukom et al., 2010). Besides, centuries-long tree-ring drought atlases are available for North America (Cook et al., 2010b), Asia (Cook et al., 2010a), Europe (Cook et al., 2015) and eastern Australia and New Zealand (Palmer et al., 2015) to study long-term hydroclimate variability. Steiger et al. (2018) produced the fist global hydroclimatic reconstructions at annual and seasonal resolutions by combining multiproxy

data with the Community Earth System Model Last Millenium Ensemble model simulations (Otto-Bliesner et al., 2016) over the last two millennia. A multi-century global reconstruction making use of observational precipitation data is still missing.

One challenge in global reconstructions is the required observation network density. To test how dense a network for a climate field reconstruction needs to be, pseudo proxy experiments can be performed. A set of such experiments were conducted by Gomez-Navarro et al. (2015) to reconstruct precipitation over Europe with three climate field reconstruction methods

(Canonical Correlation Analysis, Analog Method, Bayesian hierarchical method). They found that the skill of the precipitation reconstruction increases close to the proxy locations independently from the method used. However, further away precipitation is not reconstructed accurately. Therefore, they conclude that a dense network of proxy records, in accordance with the high spatio-temporal variability of precipitation, is essential for successful reconstructions.

Thanks to the introduction of new techniques in the field of paleoclimatology, nowadays spatially complete and physically

consistent reconstruction can be derived. Paleoclimate data assimilation provides a framework in which observational data and model simulations are optimally combined to obtain global, 3-dimensional climate fields. Using the data assimilation approach is advantageous, because by assimilating only one type of climatic variable (e.g., temperature), we can gain information from other climatic fields present in the model simulation based on their covariances. Monthly instrumental temperature and sea-level pressure data, documentary temperature data, as well as tree-ring proxy records have been successfully used with the

ensemble Kalman Fitting (EKF) method to reconstruct past climate fields (Franke et al., 2017).

In this paper, monthly precipitation information such as precipitation amounts or the number of days with precipitation in each month (wet days) are assimilated with the EKF method (Franke et al., 2017) to reconstruct monthly climate fields. Early instrumental precipitation measurements are available since the 17th century (New et al., 2001). Descriptive weather journals were kept by scholars even before the instrumental era, usually including the number of wet days. Example of such weather

diaries exist for example for Kassel (Germany) by Uranophilus Cyriandrus covering 1621–1650 (Lenke, 1960), for Zürich (Switzerland) by Wolfgang Haller between 1545–1576 (Pfister et al., 1999), or for Savanna-la-Mar (Jamaica) by Thomas Thistlewood from 1750 to 1786 (Chenoweth, 2003). In future reconstructions, these documentary records and numerous other could add valuable information to the limited instrumental measurements data to assess the natural variability of precipitation. Here, we investigate the possibility of using these new observation sources in a climate context by conducting and evaluating





a set of experiments over the 1950-2004 time period. The skill of the reconstructions with the assimilation of precipitation amount versus wet days records are compared. Precipitation amount and wet days records are assimilated together with other instrumental measurements to study their combined effect.

This paper is structured as follows: In Section 2 an overview of the EKF techniques is given and the model simulation and the observations are introduced. Section 3 describes the experimental framework and the skill metrics used for evaluation. In Section 4 the results are presented and are discussed in Section 5. We conclude how monthly precipitation information can be assimilated best in Sect. 6.

## 2  Methods

### 2.1  Ensemble Kalman fitting

To reconstruct climate fields by assimilating monthly precipitation amounts and the number of wet days per month, the ensemble Kalman fitting (EKF) data assimilation technique was used (described in detail in Bhend et al. (2012) and Franke et al. (2017)). The EKF is used in an offline manner, that is only the update step of the Kalman filter is implemented. In the update step of the EKF precomputed model states (see in Section 2.2) are updated with observational information as:

$$\overline{\boldsymbol{x}}^a = \overline{\boldsymbol{x}}^b + \mathbf{K}\left(\boldsymbol{y} - \mathbf{H}\overline{\boldsymbol{x}}^b\right) \tag{1}$$

$$\boldsymbol{x}'^a = \boldsymbol{x}'^b + \tilde{\mathbf{K}}\left(\mathbf{H}\boldsymbol{x}'^b\right) \tag{2}$$

where the analysis ensemble mean ($\overline{\boldsymbol{x}}^a$) and the deviation from it ($\boldsymbol{x}'^a$) are updated separately. $\boldsymbol{x}^b$ is the background state vector, that is provided by the model simulation. $\mathbf{H}$ is the forward operator, by which the model value is transformed to observation value. In our setup $\mathbf{H}$ is linear. $\mathbf{K}$ and $\tilde{\mathbf{K}}$ denote the Kalman gain matrix and the reduced Kalman gain matrix, which are weighting the model and observation information based on their error estimates:

$$\mathbf{K} = \mathbf{P^b H^T}\left(\mathbf{H P^b H^T} + \mathbf{R}\right)^{-1} \tag{3}$$

$$\tilde{\mathbf{K}} = \mathbf{P^b H^T}\left(\left(\sqrt{\mathbf{H P^b H^T} + \mathbf{R}}\right)^{-1}\right)^T \times \left(\sqrt{\mathbf{H P^b H^T} + \mathbf{R}} + \sqrt{\mathbf{R}}\right)^{-1} \tag{4}$$

where $\mathbf{P^b}$ is the background-error covariance matrix and $\mathbf{R}$ is the observational-error covariance matrix. In ensemble-based Kalman filter methods, $\mathbf{P^b}$ is calculated from the ensemble members (Evensen, 2003). The estimation of the observational error is discussed later in Section 2.3.2. Gaussian statistics in the errors are assumed.

The EKF is implemented serially, that is, observations are assimilated one-by-one, which makes the assimilation process computationally more efficient. The localization of $\mathbf{P^b}$ is necessary if the covariances are calculated from a much smaller ensemble size than the model dimension (discussed among others in Kepert (2009)). Therefore, $\mathbf{P^b}$ was multiplied element-wise with a correlation function to eliminate spurious correlations from $\mathbf{P^b}$. The correlation function was represented with the isotropic Gaussian function:

$$G = exp\left(-\frac{z^2}{2L^2}\right) \tag{5}$$



where $z$ denotes the distance between two grid boxes and $L$ is the length scale parameter (Gaspari and Cohn, 1999). In most of the experiments of this study, we use the localization length scale parameters determined by Franke et al. (2017). The localization length scale parameter for temperature is 1500km, for pressure is 2700 km and for precipitation is 450 km. In the case of the number of wet days, the localization length scale parameter is set to 450 km. Between two different variables always the smaller $L$ is used for localization.

## 2.2 Model simulation

The Chemical Climate Change over the Past 400 years (CCC400) model simulations serve as model background. They were produced with the ECHAM4.5 model (Roeckner et al., 2003) and have 30 ensemble members (Bhend et al., 2012). CCC400 is a forced model simulation of the atmosphere that was performed at T63 horizontal resolution with 31 layers in the vertical (for more details see Bhend et al. (2012), and Franke et al. (2017)). CCC400 was used in previous climate studies (Franke et al., 2017; Valler et al., 2019), but here an additional climate field (wet days) was added to the already existing monthly model fields. Hence, in this study the background state vector ($\boldsymbol{x}^b$) has the following monthly climate fields: temperature, precipitation, sea-level pressure, and wet days. Previous climate model evaluation studies showed that precipitation characteristics such as frequency and intensity are often over- or underestimated in the model simulations (e.g., Sun et al., 2006; Koutroulis et al., 2016). The fraction of light precipitation is overestimated by the models, that is it rains too frequently (Sun et al., 2006). Therefore, in this study the threshold for wet day was set to $\geq 1$ mm day$^{-1}$. The precomputed model states cover the period between 1601 and 2004. However, in this study we mainly focus on the period between 1950 and 2004.

## 2.3 Observation

### 2.3.1 Observation dataset

Precipitation observation data are obtained from the Global Historical Climatology Network - Daily database (GHCN-Daily v3.2.2; Menne et al., 2012). Before 1950, the number of available station data in GHCN-Daily drastically decreases in large part of the world (Menne et al., 2012). Therefore, we only use time series from the 1950–2005 period. Furthermore, to ensure homogeneous data completeness and spatial coverage, we kept only those stations that fulfill the following criteria: 1) at least 50 years of data are available; 2) the distance between two stations is at least 200 km. After the selection process, 432 stations remained (Fig 1a). In the extratropical Northern Hemisphere the station network is dense and equally distributed. However, in the Southern Hemisphere, outside Australia only very few stations data are available.

The daily precipitation sums were converted to monthly totals. For the conversion of daily precipitation sums into the number of wet days, a threshold of 1 mm was used (i. e., days with precipitation < 1 mm were not considered as a wet day).



### 2.3.2 Representativity error

Rain gauge measurements are subject to systematic, random, and representativity errors (e.g., Lopez, 2013). The representativ-
ity error, in particular, arises when comparing point measurements to area averages such as grid points in model simulations.
For precipitation, it is arguably the largest error source due to the high spatial variability of this variable.

To estimate the representativity error, we analyzed all GHCN-Daily station data over the contiguous United States and
adjacent territories (lat: $24°$-$48°$ N, lon: $126°$-$68°$ W) in the 1961–1990 period. This region has the required network density
and covers several climatic zones, which makes it a good subset to estimate global errors in monthly precipitation amount and
number of wet days.

The representativity error was estimated using the following procedure: 1) An average value for all months between 1961
and 1990 was calculated at each single grid box from the stations located within one grid box according to the resolution
of the CCC400 model simulation. 2) The spatially averaged monthly time series of each grid box were subtracted from all
precipitation and wet days series within the grid box. 3) The standard deviation of the resulting series was calculated. 4) The
median value of the standard deviation was taken as the representivity error. Figure 2 shows the distribution of the standard
deviations. The median of standard deviation of precipitation and number of wet days are 24.63 mm (equivalent to 28% error)
and 1.49 days (equivalent to 21% error), respectively.

## 3 Experiments

### 3.1 Experiment design

An important feature of our assimilation process is that only anomalies are assimilated. Both model and observation data are
transformed into anomalies, using a moving 70-year time window around the current year (this window is shorter at the edges
of the available period, from which the anomalies are calculated). Working with anomalies alleviates the problem of model
biases (see Franke et al., 2017).

If more than one stations are available within the same grid box, then the average of the observation anomalies was assim-
ilated. As in previous studies (e.g., Franke et al., 2017) the length of the assimilation window is 6 months. Therefore, in $x^b$
6 monthly fields of different variables are combined. The assimilation windows cover the period between October-March and
April-September, which were chosen due to growing seasons of trees if tree-ring proxy data get assimilated. We eventually
aim at assimilating precipitation data in combination with proxy data. Hence, the half-yearly assimilation scheme was kept the
same.

When data of 6 months are combined together in $x^b$ for one assimilation step, the assimilated observations can influence all
the variables of the 6 months. However, Valler et al. (2019) found that in the case of multiple months combined into one state
vector, the analysis improved when time localization was applied, that is observation can only affect the current month. Hence,
time localization was applied in all experiments described in this paper.





A set of experiments were conducted, which are summarized in Table 1. In our first experiment we analyze the skill of the
150 reconstruction when assimilating monthly instrumental precipitation amount in the period 1950–2004 (exp_R). In the second
experiment we assimilate the monthly number of wet days instead of the precipitation amount (exp_W). Both experiments were
repeated with an extended spatial localization distance in order to make better use of the available observations (exp_R_2L and
exp_W_2L). The length scale parameter both for precipitation amount and wet days was doubled from 450 km to 900 km, based
on calculation of decorrelation distance of precipitation in the model space. In two further experiments, we added instrumental
temperature and pressure data to the assimilated precipitation amount or to the wet days records (exp_TPR and exp_TPW,
respectively) to study the combined influences of various variables (Fig. 1b). Due to the localization of $\mathbf{P^b}$, the sequence of
assimilated observations can affect the analysis ($\boldsymbol{x}^a$) (Greybush et al., 2011). Hence, we assimilated temperature data first,
followed by pressure and precipitation (or the number of wet days). The assimilated temperature and pressure time series are
described in Franke et al. (2017). In all experiments, we used an observational error of 30% or at least 10 mm for precipitation
amount, and of 2 days for wet days. These values were chosen in agreement with our estimation of the representativity error
(Sect. 2.3.2). For temperature and pressure we applied the observational errors of $\sqrt{0.9}$ K and $\sqrt{10}$ hPa, respectively (Valler
et al., 2019).

Finally, an additional experiment was conducted to reconstruct a severe drought event in Europe, in order to demonstrate the
potential of assimilating precipitation data. Seven available stations from GHCN-Daily were used to reconstruct the extreme
drought year of 1842 (Brázdil et al., 2019, and references therein). Brázdil et al. (2019) collected several documentary data
sources from diaries through newspapers to scientific papers to analyze the severity of the drought. Besides documentary data,
instrumental measurements were also examined. The year 1842 was characterized by low water levels in many countries from
the Netherlands to western Ukraine and from Italy to Sweden affecting shipping, availability of drinking water and agriculture.
We looked how this drought event is captured in an independent reconstruction, using the Twentieth Century Reanalysis
version 3 (20CRv3; Slivinski et al., 2019). In 20CRv3 only pressure measurements are assimilated and in 1842 globally
39 observations data were used. The monthly precipitation anomaly fields of 20CRv3 are calculated from the 1836-1877
time period. Besides, we compared our exp_R experiment with a seasonal high-resolution precipitation reconstruction over
Europe (Pauling et al., 2006). Pauling et al. (2006) used various data types (instrumental, documentary, proxy) and principal
component regression techniques to reconstruct precipitation between 1500 and 2000. From their dataset, we calculated the
175 seasonal relative precipitation anomalies of 1842, dividing the absolute anomaly with the 71-year climatology, centred on 1842.
Relative precipitation anomalies were used for monthly field comparison as well.

## 3.2 Evaluation

In order to evaluate the skill of the sensitivity experiments two commonly used skill metrics — correlation and the reduction of
error (RE, Cook et al., 1994) — are calculated over the 1950–2004 time period. The CRU TS3.10 dataset (Harris et al., 2014)
was chosen as a reference dataset to evaluate the precipitation, temperature and sea-level pressure reconstructions. The CRU
TS3.10 is a gridded dataset, which is relaxed to the 1961-1990 climatology in the case of no available observation. Correlation
is calculated between the absolute values of the ensemble mean of the analysis and the reference series; it expresses the



covariability between the two series, but does not measure biases. In Section 4.1, the analyzed correlation results show the correlation differences between the correlations calculated from the analysis-reference data and from the model-reference data. We show the correlation differences because the CCC400 model simulations are transient, forcing-dependent simulations and already have skills. The RE skill metric is calculated as:

$$RE = 1 - \frac{\sum (\boldsymbol{x}_i^u - \boldsymbol{x}_i^{ref})^2}{\sum (\boldsymbol{x}_i^f - \boldsymbol{x}_i^{ref})^2} \tag{6}$$

where, $\boldsymbol{x}^u$ is the reconstruction, $\boldsymbol{x}^f$ is the model simulation, $\boldsymbol{x}^{ref}$ is the reference dataset and *i* refers to the time step. In those experiments when only precipitation amount or the number of wet days are assimilated, the RE is calculated on the anomaly level from the ensemble mean of the analysis and the ensemble mean of the model simulation and compared to the reference dataset. In the case of exp_TPR and exp_TPW experiments, the $\boldsymbol{x}^f$ is replaced with the ensemble mean of the analysis from the exp_TP experiment. The two skill metrics measure different properties, therefore the skill of the reconstruction also depends on the analyzed skill metrics (Franke et al., 2019). When for example only the number of wet days are assimilated then the temperature reconstruction is fully independent from the reference dataset. However, in other cases the skill of the reconstruction can be overestimated (see also in Franke et al., 2019). Monthly skill metrics are discussed for the exp_R experiment, while for the other experiments the seasonal skill metrics – calculated from seasonal averages (winter: October-March, summer: April-September) – are analyzed. Both seasons are discussed later but only the results of summer season are shown in the paper.

## 4 Results

### 4.1 Skill scores analysis

Assimilating monthly precipitation amount data (exp_R) led to improved precipitation skill compared to the existing correlation between the CCC400 model simulation ensemble mean and the reference dataset. The monthly correlation differences of precipitation show clear improvement (winter: Fig. S1, summer: Fig. S2). The correlation differences between the analysis and the model simulation are almost always positive in the case of temperature and sea-level pressure in all months (Fig. S1, Fig. S2). In terms of RE values, during boreal winter the monthly precipitation fields show reduced skill in the high northern latitudes and in Siberia (Fig. S1). The skill of the precipitation reconstruction gradually decreases from October to March over Siberia (Fig. S1). The negative RE values remain present in April and to a less extent in May over northern North America and Siberia; while the RE values are mainly positive throughout June and September in the Northern Hemisphere (Fig. S2). The precipitation reconstruction has mixed skill over Australia. Mostly negative RE values characterize the northern, north-western regions, while positive skill is seen in the southern and eastern parts (Fig. S2). The RE values of the winter monthly temperature fields are in general positive, except Australia (Fig. S1). However, the skill of the temperature reconstruction is negative in large parts of North America and Europe especially from June to August (Fig. S2). In the winter sea-level pressure fields, an improvement can be seen mainly over Europe and Asia (Fig. S1), while in the summer months the effect of precipitation on





the pressure fields is mixed (Fig. S2). The winter seasonal skills of exp_R are shown in the supplementary material (Fig. S3
and Fig. S4), while summer seasonal skills are shown in Fig. 3 and Fig. 4.

The assimilation of wet days (exp_W) resulted in mainly positive correlation differences of all the three variables (Fig. 3,
Fig. S3). The correlation skill of the exp_W analysis is very similar to the skill of exp_R, in which precipitation amounts
were assimilated. However, the skill of the reconstructions shows a different picture when the RE values are analyzed. The
precipitation field shows improved skill almost everywhere in summer (Fig. 4c), and in winter the RE values are negative only
in the high northern latitudes and over west and central North Asia (Fig. S4). The negative skill over Europe seen in the summer
temperature field when assimilating precipitation amounts (Fig. 4e) is diminished by using the number of wet days (Fig. 4g).
However, the negative RE values over the central Siberian region seen in summer in the exp_R experiment did not change with
the assimilation of wet days (Fig. 4d). In general, the assimilation of wet days (exp_W) has higher skill than the assimilation
of precipitation amounts (exp_R), when temperature fields are compared. The positive effect of assimilating the number of wet
days instead of precipitation amount is also evident in the RE values of the sea-level pressure fields (Fig. 4k, Fig. S4). The
sea-level pressure analysis of exp_W has positive skill over Europe in both seasons.

Doubling the correlation length scale parameter of precipitation resulted in very similar correlation skill of precipitation
(Fig. 3b, Fig. S3) as in the exp_R experiment. Increasing the correlation length scale parameter mainly positively affected
the sea-level pressure reconstruction (Fig. 3j). The same holds for the temperature correlation (Fig. 3f). While the increased
localization positively affected the correlation of the reconstructed fields, it had a more mixed impact on the other skill metric.
The RE values of precipitation decreased in the high northern latitudes, over North America and in the Mediterranean region
(Fig. 4b, Fig. S4). This negative effect of the increased localization length scale is also seen in the sea-level pressure primarily
in summer (Fig. 4j). The temperature reconstruction is affected differently in the two seasons. In winter the RE skill mainly
increased in the Northern Hemisphere (Fig. S4), while in summer larger areas are affected negatively (Fig. 4f).

The same experiment with doubled localization length scale parameter was conducted with the assimilation of the number
of wet days. Correlation coefficients of precipitation in the exp_W_2L experiment remained high (Fig. 3d, Fig. S3) similar to
what was obtained with a more strict localization scheme in the exp_W experiment. Both sea-level pressure and temperature
reconstructions benefited from the larger localization length scale, except a few grid boxes (Fig. 3h,l; Fig. S3). The precipitation
reconstruction shows similar skill to the exp_R experiment in terms of RE values. However, the increase of the localization
length scale had a positive impact on the sea-level pressure fields over Europe in both seasons (Fig.4l, Fig. S4). Over central
South America and South Africa, that were negatively affected in the exp_W experiment, the region with negative RE values
increased in the exp_W_2L experiment (Fig. S4). Similarly to the sea-level pressure field, in some regions the temperature
reconstruction became more skillful, while in others the skill decreased (Fig. 4h, Fig. S4).

In the next experiment different observation types such as temperature, sea-level pressure and precipitation amounts were
combined (Fig. 1b) as it should be done in the real application of the method in the future. In previous studies, monthly instru-
mental temperature and sea-level pressure data were already successfully assimilated (Franke et al., 2017; Valler et al., 2019).
Here we added precipitation measurements to the assimilated variables. To see the effect of assimilating precipitation data
an experiment using only temperature and sea-level pressure data was carried out over the 1950–2004 time period (exp_TP),





which resembles the original setup in Franke et al. (2017) besides not assimilating any proxy information. In the exp_TPR
experiment, first the temperature, then the sea-level pressure data and finally the precipitation amounts were assimilated. As
mentioned above, in this experiment the skill of the exp_TPR analysis is compared to the exp_TP analysis mean. Precipitation
correlations show a marked improvement over the exp_TP in both seasons (Fig. 5b, Fig. S5), as expected. Temperature and
sea-level pressure correlations are not changed in most part of the world (Fig. 5e, h; Fig. S5). The positive influence of assim-
ilating precipitation amount on the precipitation reconstruction is not only seen in the correlation but also in the RE values.
The exp_TP experiment has positive RE values mainly over Europe in both seasons (Fig. 6a, Fig. S6). Adding precipitation
amount to the assimilated observations (exp_TPR) increased the precipitation reconstruction RE skill in winter for example,
across Europe to the Urals (Fig. S6), while in summer the skill tends to be more positive from the Ural to North and East Asia
(Fig. 6b). The temperature reconstruction in terms of RE values mainly show slight improvement in winter (Fig. S6) but in
summer the skill is negatively affected by the assimilated precipitation amount in North America and Europe (Fig. 6e). The
impact of precipitation amount on the sea-level pressure reconstruction is mixed. In Europe many pressure records are available
and it is over Europe where the skill decrease the most in both season (Fig.6h, Fig. S6). Besides Europe, North America is also
negatively affected, especially in summer (Fig. 6h). The experiment was repeated with the number of wet days added instead
of precipitation amounts (exp_TPW). The correlation differences of all three variables are very similar to the differences seen
between exp_TPR and exp_TP (Fig. 5, Fig. S5). The precipitation RE values also show a similar pattern to the exp_TPR ex-
periment, but for instance over Australia in boreal winter the skill tend to be higher (Fig. S6). The summer season temperature
reconstruction of exp_TPW is in some regions more skillfull than the exp_TP experiment (Fig. 6). The sea-level pressure fields
of exp_TPW depict negative RE skill over Europe (Fig. 6i, Fig. S6).

## 4.2   Case study: 1842 drought

Seven stations in the GHCN-Daily dataset in Europe fulfilled the requirement to have data in 1842. These stations, with
their starting year in parenthesis are the following: Prague (1804), Jena (1826), Hohenpeissenberg (1781), Bologna (1813),
Genoa (1833), Mantova (1849), and Armagh (1835). Using all available data in the 71-year time window centred on 1842
the precipitation amounts and the number of wet days were converted to anomalies. We calculated the relative precipitation
anomalies for the months between April and September from exp_TP, exp_R, exp_W and 20CRv3.

In the exp_R experiment (when only precipitation amount was assimilated) the period between April and August in central
Europe is characterized with negative relative precipitation anomalies (Fig. 7b). The experiment was repeated by assimilating
the number of wet days (exp_W) instead of the precipitation amounts. The monthly relative precipitation anomaly fields are
similar to the exp_R precipitation fields, but the deviations from the climatology is smaller (Fig. 7c). However, in May the
South of France and Northern Italy are wetter than the climatology in the wet days experiment (Fig. 7c), while in the exp_R
experiment these regions are drier/closer to the climatology (Fig. 7b). We also investigated, how this drought event is repre-
sented in the exp_TP experiment when no direct precipitation data were assimilated. The relative anomalies of precipitation
indicate a dry period over Central Europe from June to August, with the largest precipitation deficit in August (Fig. 7a).





These monthly relative anomaly fields were compared to the 20CRv3 reanalysis. As mentioned above, no precipitation data were used in 20CRv3. The relative precipitation anomaly fields of 20CRv3 and exp_R show similar precipitation patterns over the central region from France to western Ukraine. A precipitation deficit from April to August and a wet September is
reproduced in 20CRv3 (Fig. 7d).

The drought over Europe in exp_R stands out even better on a seasonal time scale (June–July–August) (Fig. S7). Except northern North Europe, Spain and the southern part of France and Italy, negative relative precipitation anomalies define Europe (Fig. S7). Despite assimilating precipitation amount data from only seven stations the obtained precipitation pattern is very similar to the reconstruction made by Pauling et al. (2006) which is based on several precipitation series, documentary data
and proxy records (Fig. S7). However, there are regional differences. In their reconstruction, a larger area with more negative anomalies is present over Central Europe, while we find the strongest relative anomaly in exp_R over the north part of Central Europe. The sign of the precipitation anomaly differs over the Eastern Mediterranean region, but in exp_R no data is used from this area.

## 5    Discussion

### 5.1    Skill scores analysis

In weather forecast, there have been many attempts to make use of precipitation measurements from radar (e.g., Lopez, 2011), satellite (e.g., Lien et al., 2016), and rain gauges (e.g., Lopez, 2013). However, the usage of variables with non-Gaussian distribution — such as precipitation — is a challenging task in some data assimilation methods; therefore, different transformation techniques have been tested (e.g., Lien et al., 2016). Monthly data have the advantage of being closer to normality, as a conse-
quence of the central limit theorem. In the case of precipitation, however, this only applies to sufficiently wet climates, where several precipitation events occur each month.

Hence, the question remains whether the errors of precipitation amounts and the number of wet days are normally distributed, a fundamental assumption of the EKF. In Fig. 8, we show the outcome of a Shapiro-Wilk test for normality applied to the deviations of the ensemble members from the ensemble mean, which represent our best estimation of the background error.
On the one hand, we see that the distributions in arid climates or seasons are significantly different from a normal distribution: in these conditions the assimilation of precipitation by means of the EKF can not provide the optimal solution. On the other hand, large parts of the mid-latitudes show errors close to a normal distribution, particularly in winter. The number of wet days is less often rejected than precipitation, indicating that the former is more suitable for assimilation. This is probably one of the reasons why the skill of the reconstruction is much better in Europe when using wet days, although the climate of Southern
Europe is too dry in summer for either variable to be assimilated.

Another advantage of the number of wet days over precipitation amounts is its lower representativity error (see Sect. 2.3.2). The observation error of the number of wet days is also arguably lower, because it is less affected by undercatch. Yang et al. (1999) showed that unshielded manual rain gauges underestimate solid precipitation by 10 to 50% on average, depending on





location. The impact of the undercatch on the number of wet days is much smaller, since it only affects days with precipitation

between 1-2 mm (assuming a 50% undercatch).

Assimilating monthly precipitation information such as precipitation amounts and the number of wet days, in general, show positive improvements in all variables compared to the CCC400 model simulations, when correlation is analyzed. The RE skill of precipitation from both exp_R and exp_W experiments is negative at the high northern latitudes and over large part of Asia, especially in the winter season. As discussed earlier, precipitation observations are not error free. Prein and Gobiet

(2017) compared different gridded precipitation datasets over Europe and found large differences between them, comparable to uncertainties seen between regional climate model simulations. They defined several possible sources of observational uncertainties, for example precipitation undercatch correction or station densities. The North Asia region prior to 1957 shows the largest differences between different datasets in the Northern Hemisphere (Harris et al., 2014). Hence, it is difficult to evaluate the skill of the precipitation reconstruction from the exp_R and the exp_W experiments in this region. Using an

ensemble of observational datasets for evaluation of the reconstruction could provide a better strategy (Prein and Gobiet, 2017). The RE skill of temperature and sea-level pressure from exp_R and exp_W experiments is not always positive, which suggests that the variance of the reconstructions is different from the chosen reference series. However, using the number of wet days instead of the precipitation amounts greatly improved the RE skill of both temperature and sea-level pressure reconstructions over Europe, especially in summer.

In addition to assimilating only precipitation amounts or only the number of wet days, the effect of assimilating them in combination with other types of instrumental measurements was also tested. Correlation and RE skill metrics were calculated, using the reconstruction based on temperature and sea-level pressure data (exp_TP) as a baseline. The exp_TP experiment show a clear positive impact on the correlation values of all the three variables in both seasons. Due to the high skill of temperature and sea-level pressure reconstructions, further improvements with the assimilation of precipitation amounts or

the number of wet days are mainly seen in the precipitation field. Precipitation amount and number of wet days data have small impact on the temperature and sea-level pressure correlation skills. If only precipitation information is available from a given region, they affect more the reconstruction of the other fields; for example over North and West Australia, where no temperature observations are assimilated, precipitation information improved the skill of the temperature field in boreal winter (Fig. S4). The RE skills of all variables from exp_TPR and exp_TPW are very similar in winter, indicating that both

precipitation data perform similarly. Moreover, assimilating precipitation amount or wet day records have mainly positive effect in the regions where temperature observations are absent. However, sea-level pressure fields suggests that with the assimilation of precipitation observations the skill of the pressure reconstruction decreased in the regions of available sea-level pressure measurements. In general, precipitation amounts (exp_TPR) performs worse than wet day records (exp_TPW) in summer. The better performance of wet days in summer is expected due to their lower spatial variability. However, even wet days records

have a negative effect on the sea-level pressure RE skill, especially in Europe.



## 5.2 Case study: 1842 drought

A case study was conducted to test how well precipitation amounts and wet day records are able to reproduce a severe drought event in Europe. Only a sparse network with 7 stations provided data from 1842. In the model simulations (CCC400) the precipitation anomalies are much smaller compared to the reconstructed precipitation fields in exp_R and exp_W experiments,

indicating that with the assimilation of 7 stations this drought event is reconstructed. The reconstructed precipitation anomaly fields are very similar in the case of both precipitation amounts and the number of wet days (Fig. 7), which is promising, since in the past more wet day records are available. In the exp_TP experiment the dry event is also present, especially in August; however, the number of temperature and pressure observations which were assimilated are much higher than the seven available precipitation stations.

Based on the historical sources gathered by Brázdil et al. (2019) a significant precipitation deficit was reported in Bohemia from April to December, causing forest fire in June. Similarly, a very dry period was recorded in Prague between April and August, with extreme dryness in August. Besides the Czech Lands, several documentary data between April and August describe similar conditions in Germany, like fire in Hamburg, the lack of drinking water, low water level on the River Danube, and praying for rain. In the GHCN-Daily dataset there are observation measurements from these regions and in the exp_R and

exp_W experiments notable negative relative precipitation anomalies are apparent from April to August (Fig. 7), with August being the most severe in accordance with the documentary data. In the 20CRv3 the relative precipitation anomaly in August does not appear to be as large over this part of Europe; however, no pressure data is included from East-Central Europe. A larger precipitation deficit is depicted over Northern Europe in August in 20CRv3, which can be supported with documentary data about forest fires in Norway (Brázdil et al., 2019). Several other documentary records across Europe from France to

Transylvania describe extraordinary drought, which is reflected with varying intensity in the precipitation anomalies fields of exp_R and exp_W experiments and in the 20CRv3 reanalysis.

## 6    Conclusions

As the application of data assimilation techniques have become more widespread in the field of paleoclimatology, more and more different observational sources such as early instrumental measurements, documentary records, and various types of

proxy records have been used in the assimilation process. In this paper, new observation data sources — precipitation amounts and number of wet days — were tested in an offline ensemble-based Kalman filter framework. The experiments in which only precipitation amounts (exp_R) and only wet day records (exp_W) are assimilated performed similarly in winter, but in summer exp_W has better skill in the case off all three examined variables. Moreover, the results of the exp_W_2L experiment suggest that the localization used for wet days could be increased, by which a better use of observation data can be achieved.

In exp_TPR and exp_TPW experiments, the skill of the two reconstructions were compared to exp_TP, to examine the effect of adding precipitation data to the assimilated observations. In general, both precipitation amounts and the number of wet days had rather positive impact on the temperature reconstructions in winter, while in summer only the number of wet days had an overall positive effect on temperature. The skill metrics of sea-level pressure clearly indicate that precipitation data should





be used if pressure measurements are not available from a given region. Therefore, it might be better to limit how precipita-
tion data can update the other fields of the state vector, for example with the implementation of an asymmetric localization
function, in forthcoming experiments. The reconstructed monthly precipitation fields of the severe 1842 drought in Europe are
mostly in agreement with documentary data, showing that precipitation amounts or wet day records can be useful sources in a
paleoclimate data assimilation framework.

*Author contributions.* All authors were involved in the design of the study and contributed to writing the manuscript. VV conducted the
experiments and performed most of the analyses. The model wet days field and the assimilated precipitation amounts and wet days data were
prepared by YB. YB helped with the analysis. JF developed the original code.

*Competing interests.* The authors declare that they have no conflict of interest.

*Acknowledgements.* This research has been supported by the Swiss National Science Foundation (grant no. 162668) and the European
Commission – Horizon 2020 (grant no. 787574). The CCC400 simulation was performed at the Swiss National Supercomputing Centre
CSCS.



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




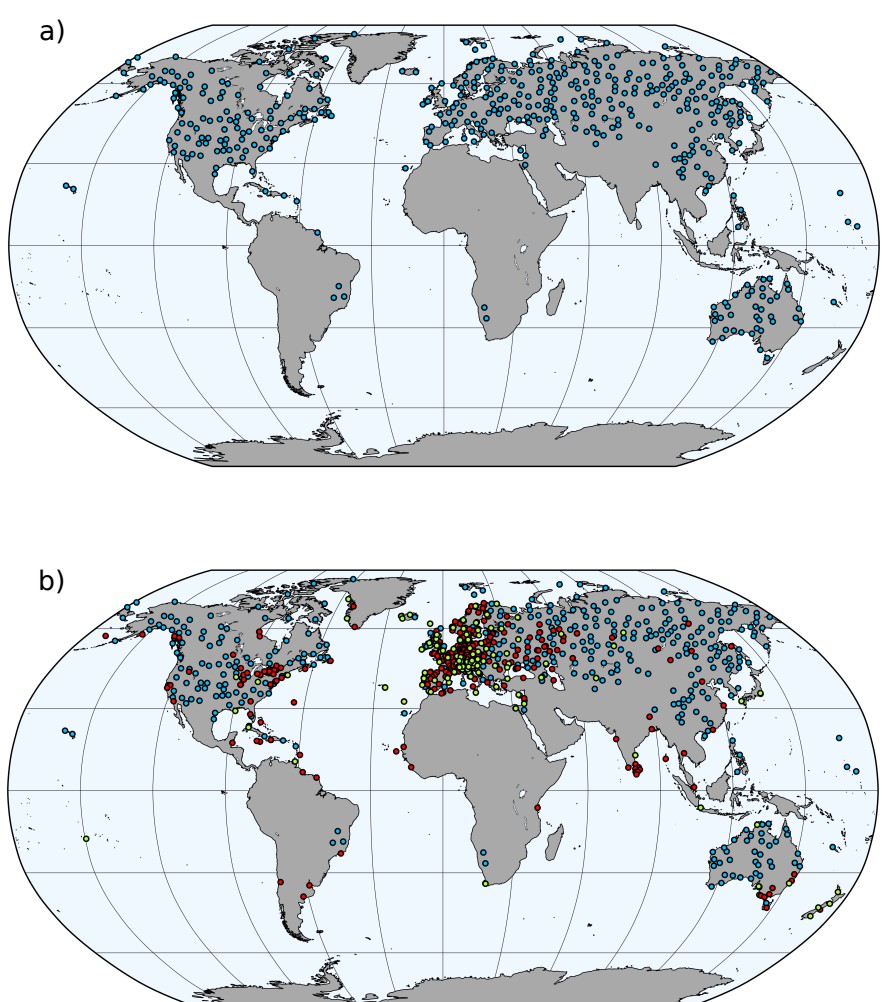

**Figure 1.** The location of precipitation observations (a) used in exp_R, exp_W, exp_R_2L, and exp_W_2L experiments. The location of instrumental precipitation (blue), temperature(red), sea-level pressure (green) observations (b) assimilated in exp_TPR and exp_TPW experiments.



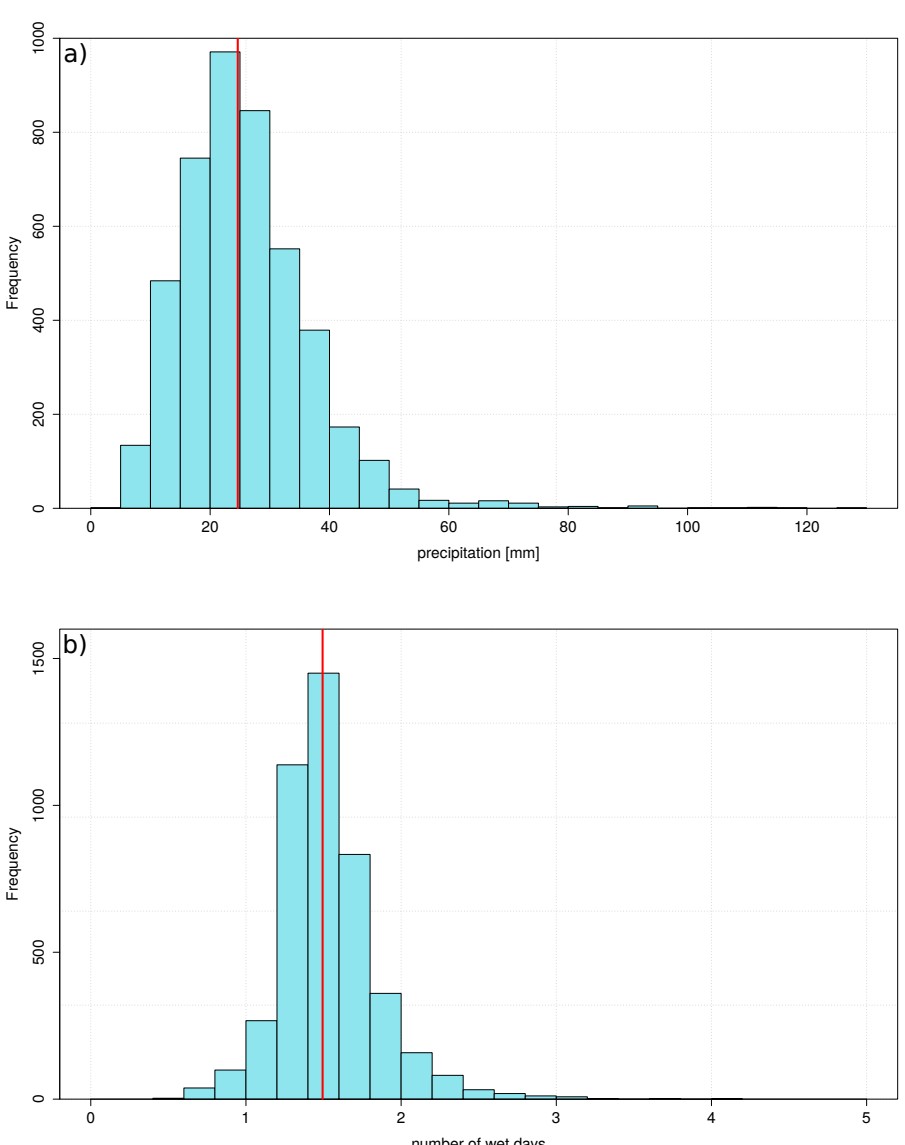

**Figure 2.** The distribution of representativity error of precipitation (a) and the number of wet days (b). The red line indicates the median.





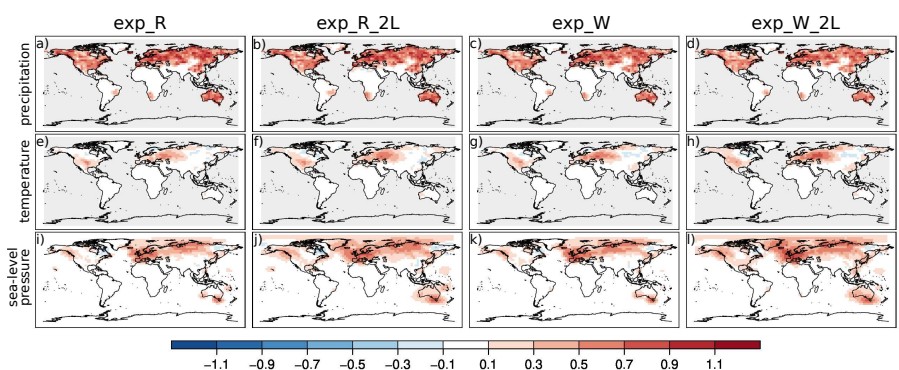

**Figure 3.** Summer season correlation differences of precipitation (a, b, c, d), of temperature (e, f, g, h), and of sea-level pressure (i, j k, l) between the analyses and the CCC400 model simulation ensemble means by assimilating only precipitation amounts (a, e, i), only precipitation amounts with doubled localization length scale parameter (b, f, j), only number of wet days (c, g, k), and only number of wet days with doubled localization length scale parameter (d, h, l).



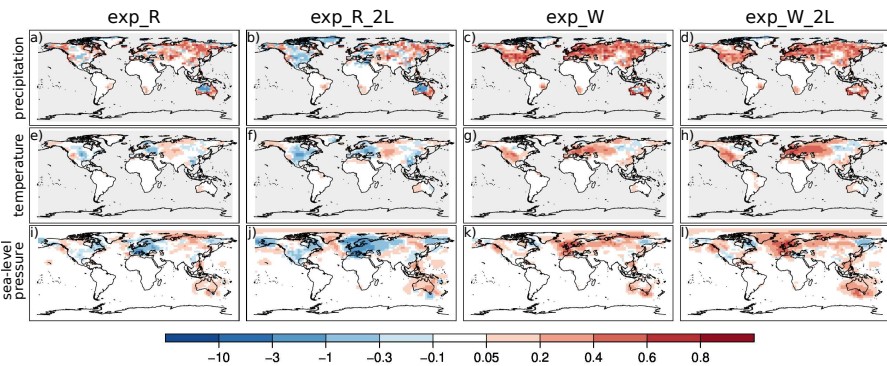

**Figure 4.** Summer season RE values of precipitation (a, b, c, d), of temperature (e, f, g, h), and of sea-level pressure (i, j k, l) by assimilating only precipitation amounts (a, e, i), only precipitation amounts with doubled localization length scale parameter (b, f, j), only number of wet days (c, g, k), and only number of wet days with doubled localization length scale parameter (d, h, l).





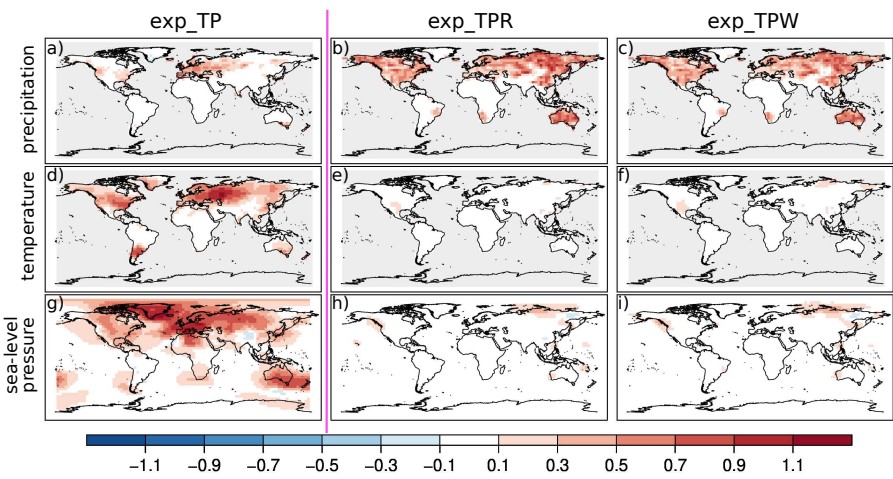

**Figure 5.** Summer season correlation differences between the analysis of exp_TP and the CCC400 model simulation ensemble means: precipitation (a), temperature (d), sea-level pressure (g). In the middle column (b, e, h) the correlation differences between the analyses ensemble mean of the exp_TPR and exp_TP are shown; while in the right column (c, f, i) the correlation differences between the exp_TPR and exp_TP analyses ensemble mean are depicted.



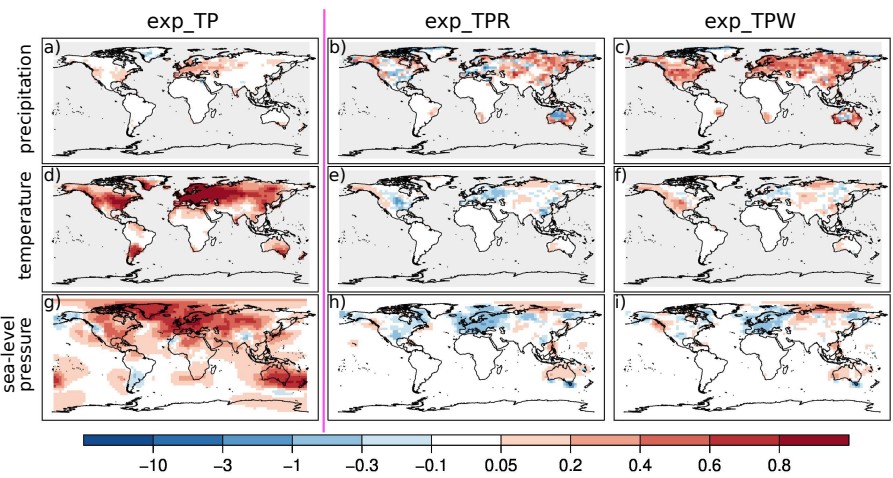

**Figure 6.** Summer season RE values of exp_TP (a, d, g), exp_TPR (b, e, h) and exp_TPW (c, f, i). $x^f$ is the ensemble mean of the CCC400 model simulations when the RE values calculated for the exp_TP experiment, while $x^f$ was replaced with the analysis mean of the exp_TP for the other two experiments.





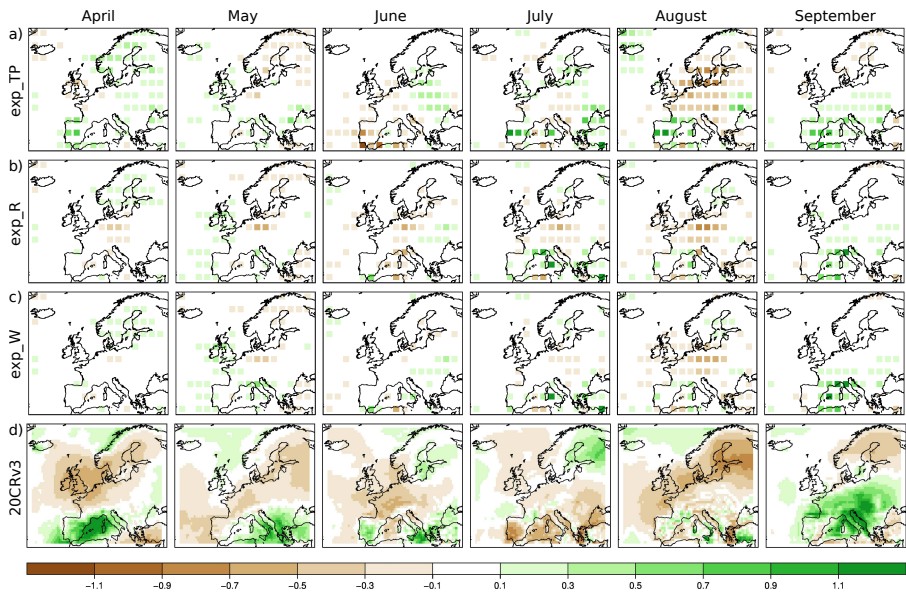

**Figure 7.** Relative precipitation anomalies over Europe in 1842. Monthly relative precipitation anomaly fields between April and September from exp_TP (a), exp_R (b), exp_W (c), and 20CRv3 (d) are depicted. Note that the anomalies are calculated from 1807-1877 period for exp_TP, exp_R, and exp_W, and from 1836-1877 for 20CRv3.

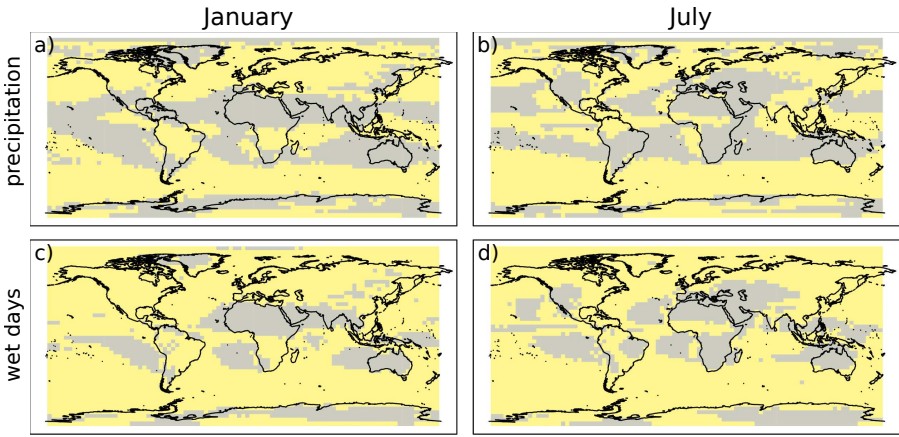

**Figure 8.** Outcome of the Shapiro-Wilk test for normality for precipitation (a,b) and number of wet days (c,d) in January (a,c) and in July (b,d). The grey area show where the null hypothesis of normality is rejected ($p < 0.05$).



**Table 1.** Summary of experiments and their setups.

| Name | Assimilated observation | Localization | Time period |
|---|---|---|---|
| exp_R | precipitation amount | L | 1841-1842, 1950-2004 |
| exp_R_2L | precipitation amount | 2L | 1950-2004 |
| exp_W | wet days | L | 1841-1842, 1950-2004 |
| exp_W_2L | wet days | 2L | 1950-2004 |
| exp_TP | temperature, pressure | L | 1841-1842, 1950-2004 |
| exp_TPR | temperature, pressure, precipitation amount | L | 1950-2004 |
| exp_TPW | temperature, pressure, wet days | L | 1950-2004 |