# Peer review of "Assimilating monthly precipitation data in a paleoclimate data assimilation framework"

_Climate of the Past, 2019_

## Referee Comment (RC1) · Anonymous Referee #1 · 6 Jan 2020

General comments

This paper discusses a new method for creating a monthly paleoclimate reconstruction by assimilating precipitation data in the form of monthly precipitation accumulation or number of wet days per month. Experiments are performed over a 55-year time period. Performance is judged relative to a separate gridded reconstruction, and results are compared with a model-only simulation and with a reconstruction created by assimilating only the conventional observations of temperature and pressure. While there are some mixed results, the overall conclusion is that this method would be worthwhile to continue pursuing. I am particularly interested in one of the final statements in the conclusions regarding limiting the effects of precipitation on assimilation, presumably to decrease the influence of precipitation when sea level pressure or temperature ob-

servations are dense, with an asymmetric localization function. Overall, the paper is clear and concise. The methods are explained well and the results are shown clearly. I have only minor scientific and technical comments.

Specific comments

1. Lines 40-43: Could you give some quantification of what a "dense network" is in this context?

2. Line 50: A short discussion of the necessary density of proxies for successful reconstruction here would help to link with the previous paragraph.

3. Line 157-158: Why did you choose this order to assimilate observations (temperature, then pressure, then precipitation)?

4. End of sections 4.1 and/or 5.1: It might be worthwhile to discuss the tradeoffs in more detail. When precipitation is assimilated, is the improvement in precipitation skill worth the loss of skill in SLP and temperature?

5. Section 4.2 and related figures: It may be useful to add the observation locations from 20CRv3 to Figure 7 so that the reader has an idea of what the observational network looks like in 1842 over Europe. In addition, it would be useful to mention that the 20CRv3 analysis is an ensemble mean, and thus comes with an ensemble spread that can provide an idea of the confidence in the reanalysis at any time and location. It could be useful to add maps showing this spread, or at least add the observation locations to Fig 7 along with a few sentences noting that the reanalysis will be less reliable in locations with no observations, than in regions of dense observations. Finally, Figures 7 and S7 might be clearer if difference fields are shown. As it is, the reader is left to judge by eye which of the experiments matches the reanalysis or reconstruction best.

Technical corrections

1. Line 34: "fist" should be "first"

2. Line 48: change to "...based on the covariances between the observed and unobserved variables."

3. Line 50: "such as" should be "in the form of"

4. Line 57: "other" should be "others"

5. line 87: "change to ...with a correlation function that decreases as distance increases"

6. line 104: maybe replace "rains" with "drizzles", since light precipitation is overestimated.

7. Line 117: replace "considered as a wet day" with "considered wet days")

8. Line 126: change to "...using the following procedure for both precipitation amounts and wet days:"

9. Line 129: replace "precipitation and wet days" with "station"

10. Line 139 should be "If more than one station is available..."

11. Line 149: "A set of experiments was conducted..."

12. Line 170: Please add: "In 20CRv3 only pressure measurements are assimilated into a model with prescribed sea surface temperatures, sea ice concentrations, and radiative forcings." Also, could you expand on what "globally 39 observations data" means? Are there 39 distinct observation locations in 1842, or are there 39 instantaneous pressure reports within all of 1842?

13. Line 172: please replace "besides" with "In addition,"

14. Line 181: Is CRU TS3.10 completely independent from the experiments or does it use some of the same data?

15. Line 183 to 185: maybe change to "In Section 4.1, we show the differences between the correlations calculated from..."

16. Line 186: change "skills" to "skill"

17. Line 189-190: change to "on the anomalies from the..."

18. Line 197: remove "season"

19. Line 229-230: Change "increased localization" to "increased localization distance" or "decreased localization", since localization itself usually refers to the method of limiting spurious correlations.

20. Lines 240-242 are unclear as written; maybe change to "The regions over central South America and South Africa that were negatively affected in the exp_W experiment show worsening skill in the exp_W_2L experiment (Fig. S4)."

21. Line 245: remove "as it should be done in the real application of the method in the future", or describe what you mean by "real application".

22. Line 249: change "besides not assimilating" to "except that it does not assimilate"

23. Line 251: add "...the skill of the exp_TPR and exp_TPW analyses are compared..."

24. Line 296: change "weather forecast" to "weather forecasting"

25. Line 298: change "in some data assimilation methods" to "for many data assimilation methods"

26. Line 299: change "normality" to "Gaussian".

27. Line 303: add a reference for the Shapiro Wilk test. Which ensemble was this test performed on?

28. Line 307: Add "The hypothesis of normality in the number of wet days..."

29. Line 310: maybe add "...to be assimilated successfully."?

30. Line 322: change "for example" to "such as"

31. Line 327: change "using the number. . ." to "assimilating the number. . ."

32. Line 335: change to "Assimilating precipitation amount or the number of wet days has a small impact. . ."

33. Line 343: add "assimilating precipitation amounts. . .performs worse than assimilating wet days. . ."

34. Figures S.1-S.2: Are the columns different months? Please label.

—————————————————————

---

## Referee Comment (RC2) · Anonymous Referee #2 · 18 Mar 2020

This is a good paper that is relatively straight-forward. I think the result about assimilating wet days vs precipitation amount is particularly useful. I only have a few suggestions for clarification and improving the presentation of the results.

l.89 Could you clarify that this is the traditional Gaspari-Cohn localization function with a specific cut-off length (i.e., it is zero beyond some specified distance)?

l.156-158 What's the justification for assimilating the observation types in this order? I would have guessed that one would assimilate the observations with the longest localization length first (i.e., pressure)?

l.172 Replace "Besides" with "In addition"

Discussion of Figs. 3 & 4 I found it difficult to assess some of the comparative state-

[Figure]

ments made about these results based on the data present in the figures. The maps give an idea of what the data values are, but unless they are compared with something like box plots then it's hard to tell which option/experiment is better or best. For example, statements are made about one choice of localization length being better than another, but I couldn't tell if that was really the case given only the maps; the distributions of the data values need to be compared more quantitatively.

Figs. S1 & S2 Can you label which columns are which months?

---

## Author Response (AR1)

Dear Chantal Camenisch,

Besides the suggested changes by the reviewers, we additionally corrected Eq. 6 and renamed the reduction of error skill score to root mean squared error skill score (based on comments of another paper of us). The number of used time series in 1842 is also corrected from seven to six.

Best regards,

Veronika Valler

**REVIEWER #1**

**General comments**

This paper discusses a new method for creating a monthly paleoclimate reconstruction by assimilating precipitation data in the form of monthly precipitation accumulation or number of wet days per month. Experiments are performed over a 55-year time period. Performance is judged relative to a separate gridded reconstruction, and results are compared with a model-only simulation and with a reconstruction created by assimilating only the conventional observations of temperature and pressure. While there are some mixed results, the overall conclusion is that this method would be worthwhile to continue pursuing. I am particularly interested in one of the final statements in the conclusions regarding limiting the effects of precipitation on assimilation, presumably to decrease the influence of precipitation when sea level pressure or temperature observations are dense, with an asymmetric localization function. Overall, the paper is clear and concise. The methods are explained well and the results are shown clearly. I have only minor scientific and technical comments.

We thank the Reviewer for the careful revision of the manuscript and for improving the English of the manuscript. We will follow his/her recommendations to clarify the specified cases.

**Specific comments**

**1. Lines 40-43: Could you give some quantification of what a "dense network" is in this context?**

Based on the correlation figures in the paper by Gómez-Navarro et al. (2015) high correlations (>0.8) are found close to the pseudoproxies which drop to values <0.2 within a few hundred kilometers with seasonal dependency.

The localization length scale parameter of precipitation (450 km) that is used in this study and was previously calculated by Franke et al. (2017) is in agreement with the findings of Gómez-Navarro et al. (2015).

We added: They found that the skill of the precipitation reconstruction increases close to the proxy locations (correlations > 0.8):independently from the method used. However within a few hundred kilometers correlation values drop below 0.2 (with seasonal dependency), implying:precipitation is not reconstructed accurately anymore at these distances.

**2. Line 50: A short discussion of the necessary density of proxies for successful reconstruction here would help to link with the previous paragraph.**

The EKF400 atmospheric paleo-reanalysis (EKF400; Franke et al., 2017) predominantly shows skill over the Northern Hemisphere land areas, in the vicinity (up to a few thousand

kilometers) of assimilated data. However, in regions where no observations are available the reconstructions are identical to the model simulations.

We added: In the EKF400 atmospheric paleo-reanalysis skillful temperature and sea-level pressure reconstruction can be expected a few thousand kms away from observations, whereas precipitation fields show limited skill (Franke et al., 2017). Reasons are the high spatial heterogeneity of precipitation and that no precipitation data has been assimilated.

**3. Line 157-158: Why did you choose this order to assimilate observations (temperature, then pressure, then precipitation)?**

The reviewer is correct because the order in which the observations are assimilated may influence the results in the case of serial assimilation with localization. We kept the same assimilation order used in previous reconstruction (Franke et al, 2017), that is assimilating temperature data first then pressure measurements. We assimilated precipitation data last due to the bigger uncertainties characterizing precipitation measurements. Precipitation data also affect the reconstruction within a smaller region. A similar procedure was applied in the generation of the 20CRv3 (Slivinski et al., 2019).

We added: We kept the same assimilation order used in previous reconstruction (Franke et al., 2017), that is assimilating temperature data first then pressure measurements. Precipitation data were assimilated last due to the bigger uncertainties characterizing precipitation measurements.

**4. End of sections 4.1 and/or 5.1: It might be worthwhile to discuss the tradeoffs in more detail. When precipitation is assimilated, is the improvement in precipitation skill worth the loss of skill in SLP and temperature?**

We agree with the reviewer that it is an important question. Ideally we would like to avoid or minimize the negative effect of assimilating precipitation information and our results suggest that assimilating the number of wet days is preferable compared to assimilating precipitation amounts, especially in summer. In future experiments, in order to reduce the negative impact of assimilating precipitation information on other reconstructed fields (e.g., sea-level pressure), the effect of ignoring/limiting its cross-covariance updates will be tested.

We made a new figure (Fig. S08) and added at the end of Section 4.1: To further assess the impacts of assimilating precipitation data in regions with dense observations such as Europe, the distribution of skill metrics were analyzed. The reconstructed temperature fields of the three experiments have very similar skill over Europe (Fig. S8). Over Europe a loss of skill in sea-level pressure and temperature is seen for the RMSESS in both seasons, especially in summer, when skill of exp_TPR and exp_TPW are compared to exp_TP (Fig. 6). If the RMSESS is calculated for all three experiments using the model simulation as $x_f$, the distributions over Europe indicate that the skill loss in sea-level pressure is smaller than the gain in precipitation for exp_TPR and exp_TPW (Fig. S8).

**5. Section 4.2 and related figures: It may be useful to add the observation locations from 20CRv3 to Figure 7 so that the reader has an idea of what the observational network looks like in 1842 over Europe.**

We added the location of available observations to Figure 7 in the first column, the 6 time series from GHCN and the observations from ISPDv4.7 used in 20CRv3.

**In addition, it would be useful to mention that the 20CRv3 analysis is an ensemble mean, and thus comes with an ensemble spread that can provide an idea of the confidence in the reanalysis at any time and location. It could be useful to add maps showing this spread, or at least add the observation locations to Fig 7 along with a few sentences noting that the reanalysis will be less reliable in locations with no observations, than in regions of dense observations.**

Thank you for the recommendation. We extended the description of 20CRv3 as:
In 20CRv3 only pressure measurements are assimilated into a model with prescribed sea surface temperatures, sea ice concentrations, and radiative forcings. The 20CRv3 reanalysis has 80 ensemble members which provide an idea of the confidence in the reanalysis at any time and location. In 1842 data are available from 39 distinct locations, where the reanalysis is more relaible compared to regions with no available observations. The monthly precipitation anomaly fields of 20CRv3 are calculated from the 1836-1877 time period. The presented results compare the 20CRv3 ensemble mean with the ensemble mean of the above described experiments.

**Finally, Figures 7 and S7 might be clearer if difference fields are shown. As it is, the reader is left to judge by eye which of the experiments matches the reanalysis or reconstruction best.**

We added the additional difference figures: Fig. S09 and Fig. S10.

**Technical corrections**

The suggested technical corrections were applied.

1. Line 34: "fist" should be "first"

2. Line 48: change to "…based on the covariances between the observed and unobserved variables."

3. Line 50: "such as" should be "in the form of"

4. Line 57: "other" should be "others"

5. line 87: "change to …with a correlation function that decreases as distance increases"

6. line 104: maybe replace "rains" with "drizzles", since light precipitation is overestimated.

7. Line 117: replace "considered as a wet day" with "considered wet days")

8. Line 126: change to "…using the following procedure for both precipitation amounts and wet days:"

9. Line 129: replace "precipitation and wet days" with "station"

10. Line 139 should be "If more than one station is available…"

11. Line 149: "A set of experiments was conducted…"

12. Line 170: Please add: "In 20CRv3 only pressure measurements are assimilated into a model with prescribed sea surface temperatures, sea ice concentrations, and radiative forcings."
Also, could you expand on what "globally 39 observations data" means? Are there 39 distinct observation locations in 1842, or are there 39 instantaneous pressure reports within all of 1842? D

A sentence concerning assimilated observations in 20CRv3 was added: In 1842 data are available from 39 distinct locations.

13. Line 172: please replace "besides" with "In addition,"

14. Line 181: Is CRU TS3.10 completely independent from the experiments or does it use some of the same data?
The CRU TS3.10 dataset is not completely independent from the experiments, since there is an overlap between e.g., the assimilated temperature data and the temperature data used in the CRU TS3.10 dataset. However, for instance when only the number of wet days are assimilated the reconstructed temperature field is completely independent from the CRU TS3.10 temperature field.

15. Line 183 to 185: maybe change to "In Section 4.1, we show the differences between the correlations calculated from…"

16. Line 186: change "skills" to "skill"

17. Line 189-190: change to "on the anomalies from the…"

18. Line 197: remove "season"

19. Line 229-230: Change "increased localization" to "increased localization distance" or "decreased localization", since localization itself usually refers to the method of limiting spurious correlations.

20. Lines 240-242 are unclear as written; maybe change to "The regions over central South America and South Africa that were negatively affected in the exp_W experiment show worsening skill in the exp_W_2L experiment (Fig. S4)."

21. Line 245: remove "as it should be done in the real application of the method in the future", or describe what you mean by "real application".

22. Line 249: change "besides not assimilating" to "except that it does not assimilate"

23. Line 251: add "…the skill of the exp_TPR and exp_TPW analyses are compared…"

In line 151, we already mention that both experiments are compare to exp_TP. "In the case of exp_TPR and exp_TPW experiments, the xf is replaced with the ensemble mean of the analysis from the exp_TP experiment."

We added it again when the discussion of exp_TPW starts in line 262: "The experiment was repeated with the number of wet days added instead of precipitation amounts (exp_TPW)" and similarly as a reference the exp_TP experiment was used.

24. Line 296: change "weather forecast" to "weather forecasting"

25. Line 298: change "in some data assimilation methods" to "for many data assimilation methods"

26. Line 299: change "normality" to "Gaussian".

27. Line 303: add a reference for the Shapiro Wilk test. Which ensemble was this test performed on?

We added the reference: DS Wilks (2011) - Statistical Methods in the Atmospheric Sciences It was performed on the model ensemble.

28. Line 307: Add "The hypothesis of normality in the number of wet days…"

29. Line 310: maybe add "…to be assimilated successfully."?

30. Line 322: change "for example" to "such as"

31. Line 327: change "using the number…" to "assimilating the number…"

32. Line 335: change to "Assimilating precipitation amount or the number of wet days has a small impact…"

33. Line 343: add "assimilating precipitation amounts… performs worse than assimilating wet days…"

34. Figures S.1-S.2: Are the columns different months? Please label.

We added the labels.

**REVIEWER#2**

**This is a good paper that is relatively straight-forward. I think the result about assimilating wet days vs precipitation amount is particularly useful. I only have a few suggestions for clarification and improving the presentation of the results.**

We thank the Reviewer for his/her positive feedback and for the recommendations where to improve the manuscript.

**l.89 Could you clarify that this is the traditional Gaspari-Cohn localization function with a specific cut-off length (i.e., it is zero beyond some specified distance)?**

We implemented the Gaussian localization function as described by Gaspari and Cohn (1999), but without compact support. The elements of the localization functions are only getting close to zero, but were not set to exactly zero beyond a specified distance.

We added: Therefore, Pb was multiplied elementwise with a correlation functio that decreases as distance increases.

**l.156-158 What's the justification for assimilating the observation types in this order? I would have guessed that one would assimilate the observations with the longest localization length first (i.e., pressure)?**

We agree with the Reviewer that the order of assimilated observations can have an influence on the result. We kept the assimilation order as was used in previous reconstructions (Franke et al., 2017). However, further testing could reveal more optimal assimilation order.

We added: We kept the same assimilation order used in previous reconstruction (Franke et al., 2017), that is  assimilating temperature data first then pressure measurements. Precipitation data were assimilated last due to the bigger uncertainties characterizing precipitation measurements.

**l.172 Replace "Besides" with "In addition"**

We have replaced it.

**Discussion of Figs. 3 & 4 I found it difficult to assess some of the comparative statements made about these results based on the data present in the figures. The maps give an idea of what the data values are, but unless they are compared with something like box plots then it's hard to tell which option/experiment is better or best. For example, statements are made about one choice of localization length being better than another, but I couldn't tell if that was really the case given only the maps; the distributions of the data values need to be compared more quantitatively.**

Thank you for the suggestion. We added a new figure summarizing the skill of the experiments as box plots over the extratropical Northern Hemisphere (Fig. S05).

**Figs. S1 & S2 Can you label which columns are which months?**

We added additional labels to the figures.

[revised manuscript text omitted]